# Risk of Second Primary Malignancies in Melanoma Survivors: A Population-Based Study

**DOI:** 10.3390/cancers15113056

**Published:** 2023-06-05

**Authors:** Javier Antoñanzas, Ana Morello-Vicente, Gloria Maria Garnacho-Saucedo, Pedro Redondo, Leyre Aguado-Gil, Rafael Salido-Vallejo

**Affiliations:** 1Dermatology Department, University Clinic of Navarra, 31008 Pamplona, Spain; jantonanzas@unav.es (J.A.); amorellovic@unav.es (A.M.-V.); predondo@unav.es (P.R.); laguado@unav.es (L.A.-G.); 2Dermatology Department, Reina Sofía University Hospital, 14004 Cordoba, Spain; gloriagarnacho@gmail.com

**Keywords:** melanoma, second primary neoplasms, risk factors, prognosis, survival

## Abstract

**Simple Summary:**

This study evaluates the occurrence of second primary neoplasms (SPNs) in individuals with a history of melanoma (MM) and identify factors that increase the risk in our population. A prospective cohort study was conducted, involving 529 MM survivors from January 2005 to August 2021. Among the 529 patients, 89 were diagnosed with SPNs, with 62 being skin tumors and 37 being solid organ tumors. The estimated probability of developing SPNs after MM diagnosis was found to increase over time, reaching 4.1% at 1 year, 11% at 5 years, and 19% at 10 years. Several factors were significantly associated with a higher risk of SPNs, including older age, primary MM location on the face or neck, and the histologic subtype of lentigo maligna MM. We conclude that individuals with primary MM located on the face and neck, as well as those with the histological subtype of lentigo maligna-MM, have a higher risk of developing SPNs. Age also independently influences the risk. Understanding these risk factors can assist in developing MM guidelines that provide specific follow-up recommendations for individuals at the highest risk.

**Abstract:**

(1) Introduction: The association between melanoma (MM) and the occurrence of second primary neoplasms (SPNs) has been extensively studied, with reported incidence rates ranging from 1.5% to 20%. This study aims to evaluate the occurrence of SPNs in patients with a history of primary MM and to describe the factors that make the risk higher in our population. (2) Material and Methods: We conducted a prospective cohort study and calculated the incidence rates and relative risks (RR) for the development of different SPNs in 529 MM survivors from 1 January 2005 to 1 August 2021. Survival and mortality rates were obtained, and the Cox proportional hazards model was used to determine the demographic and MM-related factors that influence the overall risk. (3) Results: Among the 529 patients included, 89 were diagnosed with SPNs (29 prior to MM diagnosis, 11 synchronous, and 49 after MM), resulting in 62 skin tumors and 37 solid organ tumors. The estimated probability of developing SPNs after MM diagnosis was 4.1% at 1 year, 11% at 5 years, and 19% at 10 years. Older age, primary MM location on the face or neck, and histologic subtype of lentigo maligna mm were significantly associated with a higher risk of SPNs. (4) Conclusions: In our population, the risk of developing SPNs was higher in patients with primary MM located on the face and neck and with the histological subtype of lentigo maligna-MM. Age also independently influences the risk. Understanding these hazard factors can aid in the development of MM guidelines with specific follow-up recommendations for individuals with the highest risk.

## 1. Introduction

The incidence of MM has been increasing rapidly in recent decades, particularly in New Zealand, Australia, the United States and Northern Europe [1]. However, countries in the Mediterranean basin, such as Spain, have lower incidence rates (between 10 and 13 cases per 100,000 inhabitants per year) compared to more northern regions such as Sweden. The incidence rates show an inverse behavior as we move closer to the Equator, which could be attributed to the shorter phototype of the Nordic population and the intermittent but intense sun exposure during holidays [2].

The increase in both the incidence and survival rates of MM patients can be attributed to early diagnosis and therapeutic advancements, resulting in a larger number of patients requiring extended observation periods. During this follow-up, there is an elevated risk of developing a subsequent primary neoplasm, as described in the literature, with reported rates ranging from less than 1% to over 10% at 5 years, with an overall probability as high as 25% [3]. Additionally, patients with a history of MM have been associated with various other tumors, including non-melanoma skin cancer (NMSC), non-Hodgkin lymphomas (NHL), kidney, pancreas, bladder, colon, endometrium, brain, bone, salivary gland, prostate, breast cancer and chronic lymphocytic leukemia (CLL) [4,5,6,7,8,9,10,11]. Interestingly, lower incidences have been documented for pharynx, hypopharynx, esophagus, liver, larynx, lung and bronchus, cervix uteri, nonlymphocytic leukemia, myeloid and monocytic leukemia [1,12,13,14,15,16,17,18,19,20].

The risk of SPNs in MM survivors carries prognostic implications. A Swedish study covering the period from 1958 to 2015, with a median follow-up of 8 years, reported that 74.2% of patients without SPNs died from MM, while among those diagnosed with SPNs, the first MM was the cause of death for only 24.5% of patients, and 43.1% died from SPNs [21].

Nevertheless, there is controversy surrounding the published data, and the potential role of confounding factors, particularly surveillance bias, has been suggested. Understanding the risk patterns for developing SPNs in MM patients can have implications for both research and clinical practice, including cancer screening and the development of specific MM follow-up guidelines [1,3].

The objective of this study was to assess the estimated risk of SPNs following a diagnosis of MM in our population and to describe the clinical factors associated with SPNs development. The knowledge of these risk variables would make it possible to establish follow-up protocols with specific recommendations for individuals more likely to present SPNs.

## 2. Materials and Methods

### 2.1. Study Population

A longitudinal prospective follow-up study was conducted at the Reina Sofía University Hospital in Córdoba, Spain, between January 2005 and August 2021. A total of 529 patients with histologically confirmed primary MM were enrolled based on the ICD-10 classification. Patients with xeroderma pigmentosum, MM metastases without an associated primary tumor, and those with records of neoplasms prior to the MM diagnosis were excluded. Patients treated with BRAF and MEK inhibitors or immunotherapy were also excluded due to an increased risk of skin and digestive tumors, respectively.

Finally, SPNs occurring within one month after MM diagnosis were classified as synchronous and censored.

### 2.2. Data Collection

Patient information was collected by searching the databases of the Dermatology and Pathology Departments at the Reina Sofía University Hospital. The study was approved by the institutional review board of the Centre. Demographic characteristics such as age and sex, as well as tumor features including date of diagnosis, clinical subtype, location, Breslow index, Clark level, stage (according to TNM classification), adjuvant treatment received, date of loss to follow-up, date of death (in case of death), and presence or absence of a SPN, were considered. For patients with SPNs, the date of diagnosis, histological type, and location were also collected. Only SPNs with histologically confirmed malignant behavior were included. All patients were evaluated and managed based on decisions made by the multidisciplinary skin cancer board committee of the hospital.

### 2.3. Statistical Analysis

Categorical or dichotomous variables such as sex, tumor clinical subtype and location or treatment received were expressed as percentages, and the χ2 test and Fisher’s exact test were used for comparisons, whereas continuous variables including age and the Breslow index were expressed as means ± standard deviation (SD) and Student’s *t*-test was used to analyze differences between groups.

Prior to the survival analysis, a descriptive analysis was conducted for the variables included in the study, considering patient characteristics and malignant tumors detected during the follow-up. The age at MM diagnosis was qualitatively recoded into two categories: under 50 years of age and over 50 years of age, based on previous reports indicating that 50% of MM cases occur before the age of 50 [19]. The MM stage was also categorized according to the hospital protocol: Stages 0 and Ia (localized disease), Ib and IIa (regional disease), IIb, IIc, and III (advanced disease), and IV (disseminated disease).

The period at risk for the occurrence of SPNs was calculated from the date of primary MM diagnosis to the date of the last review, date of death, or date of SPNs development. SPNs were classified as cutaneous (including MM, basal cell carcinoma (BCC), and squamous cell carcinoma (SCC)) or solid (non-skin and non-hematologic tumors).

Survival curves were generated using the Kaplan–Meier survival test method in the total patient group, as well as stratified for men and women, patients over and under 50 years of age, and different locations of MM. The Log-rank test was performed to assess differences in survival between groups, and a univariate analysis was conducted to determine prognostic variables.

Finally, to calculate the estimated RR, a multivariate analysis was also performed using the Cox proportional hazards model, which included the significant variables obtained in the previously performed univariate analysis. To test the true value of the parameter based on the sample estimate, we used the Wald statistic test, and variables with a *p* > 0.15 (methodical selection procedure) were eliminated from the model. The likelihood ratio test between the major and minor model was performed and the linearity of the continuous variables elicited in the model was verified using the Box–Tidwell test. The possible interactions between the variables were also analyzed, considering variables as confounding factors if there was a coefficient percentage of change greater than 15%. The likelihood ratios between the final model and another that only contained the constant and the proportional hazards condition were checked using the graphic method of normality of the survival function. The dfbeta values were considered as a diagnostic test for extreme cases, and the graphical representation of the mantingale residuals and partial residuals was used to assess the goodness of fit. The significance level was established for *p* values < 0.05. All the statistical analysis were performed using IBM SPSS Statistics for Windows version 25.0 (IBM Corp, Armonk, NY, USA, 2017).

## 3. Results

The cohort consisted of 529 patients diagnosed with MM between 1 January 2005 and 1 August 2021. It included 240 men (45.4%) and 289 women (54.6%), with a mean age at MM diagnosis of 60 years (range 11–94 years) and a mean follow-up time of 3.8 years (45.4 months). Table 1 presents the characteristics of the MM cases in the study population.

Among the 529 patients with confirmed MM, 89 were diagnosed with a SPN. Of these, 29 were diagnosed prior to MM, 11 were synchronous tumors, and the remaining 49 developed the second tumor after MM. The latter group, representing 11.3% of the sample, was included in the survival analysis. Among the 89 patients with confirmed SPNs (Table 2), 62 had skin tumors (14 MM and 48 non-melanoma skin cancers [NMSC]), and 37 had solid tumors (9 breast, 11 gastrointestinal, 4 larynx, 5 prostate, and 8 from other locations).

A total of 49 patients developed 54 tumors after the diagnosis of MM. Of these, 34 (62.9%) were skin tumors and 20 were solid organ tumors (37.1%). Among skin tumors, second primary MM accounted for 35.3%, NMSC (BCC and SCC) accounted for 58.8%, and other skin tumors accounted for 5.9%. For solid tumors, there were three breast carcinomas (15%), six digestive tract tumors (30%), three prostate carcinomas (15%), one laryngeal carcinoma (5%), and seven tumors from other locations (35%). The estimated cumulative incidence rate for any SPN was 4.1% at 1 year, 11% at 5 years, and 19% at 10 years (Figure 1, Table 3).

Age older than 50 at MM diagnosis was associated with an increased risk of developing a SPN (Figure 2) (RR = 4.05; 95% CI, 1.8–9.1), while gender was not associated in our cohort (men vs. women: RR = 0.6; 95% CI, 0.34–1.05).

In the multiple regression model, the variables gender, Breslow index, Clark level, and MM stage were eliminated (Likelihood Ratio Test, G = 6.67 NS, Gl = 8). Age and Breslow index showed a linear scale, and all possible interactions were assessed but found to be non-significant.

The results of the univariate and multivariate analyses using the Cox regression model are presented in Table 4. Age, location of the primary MM, and histological lentigo maligna mm subtype were statistically significant factors associated with the risk of developing a SPN. Age increased the probability of developing a SPN by 1.05 times (95% CI, 1.03–1.07) for each year after MM diagnosis. Significant differences were observed in both the univariate and multivariate analyses for patients with MM located in the head and neck compared to those located in the trunk, upper extremities, and lower extremities (RR = 0.24; RR = 0.13; RR = 0.13) (Figure 3). Patients diagnosed with the histological subtype lentigo maligna-MM had a 3.33 times higher risk of developing a SPN compared to patients diagnosed with superficial spreading-MM.

## 4. Discussion

In our population, patients who were older than 50 years, had a history of lentigo maligna-MM, and with MMs located in the head and neck region exhibited the highest risk for SPNs. Therefore, more frequent and comprehensive monitoring for this specific group was strongly recommend.

Among the 529 patients included in our cohort, the cumulative probability of developing a SPN was estimated to be 11% at 5 years, which contrasts with previously reported data showing a rate of 5% at 5 years, indicating a more than 50% increased risk [5,7,14,22]. Thus, additional factors expressed in our study population may have contributed to the higher risk of SPNs. It is important to note that our cohort had a higher proportion of patients over 50 years old, and age older than 50 at MM diagnosis was associated with a 4.05 higher risk of developing a SPN. This finding is consistent with previous studies that have also identified age as a risk factor for SPNs [9,20,22].

It is important to acknowledge that younger patients have a longer life expectancy and, therefore, a potentially greater observation period, which can artificially increase the apparent risk for SPNs. In this context, the relatively low mean age of our cohort “less than 50 years” and the relatively short follow-up period (average of 5.1 years) may have contributed to the lower number of observed SPNs in this group. However, it is important to note that with a longer follow-up duration, there is a possibility that a higher rate of neoplasms could emerge.

Some studies have reported that females have better survival rates in primary MM compared to males, and this difference has also been suggested to apply to SPNs. It has been proposed that the survival advantage in women may be attributed to enhanced immunoreactivity, as evidenced by an overall excess of autoimmune diseases [21]. However, in our series, we did not observe the same gender-related survival advantage.

The increased risk of developing a SPN after the diagnosis and treatment of primary MM has been extensively documented by several studies. However, it is important to consider the potential impact of surveillance bias in these findings. Patients diagnosed with MM often undergo more frequent medical visits, imaging, and laboratory tests, and they may be more vigilant in reporting any symptoms to their doctors, leading to increased medical explorations in the years following the MM diagnosis [23,24]. This heightened surveillance may result in the detection of additional neoplasms that might have otherwise gone unnoticed. To evaluate the presence of surveillance bias, some studies have specifically examined whether the risk of developing a SPN is limited to the early years after MM diagnosis. These studies have shown that the risks of certain cancers, such as skin MM, eye and orbit MM, prostate cancer, soft tissue cancer, salivary gland cancer, and bone and joint cancers remain elevated throughout the entire study period. On the other hand, risks for cancers such as larynx and cervix uteri were significantly reduced, suggesting no surveillance bias [1,25,26]. However, for certain tumors including thyroid cancer, non-Hodgkin lymphoma (NHL), and chronic lymphocytic leukemia (CLL), the risks were significantly elevated only during the 2 to 11 months following MM diagnosis but not throughout the entire study period, indicating the possible presence of a bias [1]. These findings highlight the complexity of interpreting the observed risks of SPNs in MM survivors and the need to consider the potential influence of surveillance bias when evaluating the results of such studies.

In our cohort, the most commonly observed skin tumors among non-melanoma skin cancers (NMSC) were basal cell carcinoma (BCC), accounting for 85% of the NMSC cases detected, followed by melanoma (MM). Among solid tumors, the most frequent were prostate, breast, and digestive tract cancers, with stomach tumors being the most prevalent and accounting for 50% of the solid tumors identified. Interestingly, the development of solid tumors required a longer follow-up period compared to skin tumors (55.8 months versus 32.8 months), suggesting that the occurrence of solid tumors was not artificially elevated due to excessive testing for MM in the initial years following MM diagnosis. This finding also indicates that closer observation, particularly in the early stages, is advisable for the detection of skin tumors.

In our study, we did not observe a significant overall increase in the risk of SPNs among patients with a history of MM when compared to the risk in the general population. The mean age of our patients was 60 years, and we observed a prevalence of 16.8% for SPNs, with 11% of them occurring after the diagnosis of MM. These results were very similar to the risk of developing any neoplasm in Spain at that age, which is 15%.

Furthermore, when excluding skin tumors, the incidence of the most commonly observed solid tumors, such as gastric and breast cancers, exhibited similar or even lower rates compared to the national average (2% vs. 1% and 2% vs. 8%, respectively). Thus, it is important to note that these findings do not completely eliminate the potential risk of long-term surveillance bias. Moreover, it can be suggested that the solid SPNs identified in our series could be detected due to the close follow-up of patients with a history of MM, rather than indicating an overall increased risk in this population. Consequently, solid SPNs were identified at earlier stages with all the tumors being discovered at stages I and II, with no cases of distant disease.

Nevertheless, we consider it important to emphasize that within our study, a significantly higher risk of SPNs was observed among patients older than 50 years, with a history of lentigo maligna-MM, and with MMs located in the head and neck region. Thus, we conclude that this specific group have a real higher risk and should receive more frequent and thorough monitoring.

The underlying association between a MM medical history and the risk of developing SPNs in MM survivors has been explored, and various theories have been proposed.

Various pathogenetic mechanisms, including genetic, environmental, treatment-related, and shared risk factors have been considered. For example, mutations in the BRAF oncogene, commonly found in MM, have also been observed in other cancers. Environmental factors, such as exposure to ultraviolet radiation (UVR) from sunlight, are other well-established contributors and additional factors such as tobacco use, occupational exposures, and socioeconomic status (SES) can also increase the risk. Lasty, treatment-related factors, such as chemotherapy or radiotherapy used for primary MM treatment, may increase the risk of developing SPNs, and their use in MM has been linked to the subsequent development of non-Hodgkin lymphoma (NHL), Hodgkin’s disease, and breast carcinoma [19,20,26,27]. However, in our cohort, exposure to chemotherapy treatments did not emerge as a significant risk factor. Among the 57 patients who received adjuvant chemotherapy, only 1.8% developed an SPN, while 98.2% completed the follow-up without any additional tumors detected. These differences were found to be statistically significant (*p* = 0.047).

Relative to its genetic origin, as commented before, mutations in the BRAF oncogene are found in approximately 50% of MM, but they are also observed in other cancers such as hairy cell leukemia, papillary thyroid cancers, colorectal cancer, liver cancer, brain cancer, lung cancer, soft tissue cancer, ovarian cancer, and breast cancer, with varying rates ranging from 2% to 100% [28]. However, there is no established association between BRAF positivity in previous MM and the development of SPNs, possibly due to the somatic nature of the mutation rather than being inherited or germline [28]. On the other hand, germline mutations in BRCA2 or CDKN2A have been suggested as explanations for the link between MM and breast cancer, and CDKN2A mutations have also been proposed as a cause of the increased risk of pancreatic cancer. Additionally, TSPY gene mutations have been observed in both MM and prostate cancer [28].

Considering environmental factors, SES may also play a role and contribute to the variation in risk factors among different populations [1]. For instance, MM and other tumors such as breast cancer, prostate cancer, NHL, brain cancer, and salivary gland cancer have been reported to be more prevalent among individuals with higher SES. Conversely, a reduced risk of larynx, cervix, and esophagus cancers has been also observed [25]. Lower SES populations often have limited access to surveillance programs and are more likely to be exposed to toxins such as tobacco, leading to an increased risk of developing SPNs. Regarding the increased risk of developing a second MM as an SPNs, environmental factors also play a role, with UVR exposure being the primary factor [4,9,12,18]. MM can arise not only from melanocytes in the skin but also from melanocytes in the eye, mucosa, and leptomeninges, which account for 10% of all MM diagnoses. These non-cutaneous MM tend to be more aggressive and associated with poorer outcomes, potentially due to delays in diagnosis. Reports have shown that patients with a previous MM are more likely to develop ocular, oral, and vaginal/exocervical MM compared to the general population. In fact, some dermatologists recommend periodic ophthalmologic, dental, and gynecologic examinations for patients with a history of MM [29].

In our study, the location of the primary MM on the face and neck showed a statistically significant association with the subsequent development of SPNs compared to other locations. In the final multivariate model, which included age and histological type of MM as covariates, patients with a MM located on the trunk had a 4.16 times lower risk of developing SPNs compared to those with a primary MM on the face or neck. Similarly, patients with primary MM on the upper and lower limbs had 7.69- and 5-times lower risk, respectively. The increased risk observed in patients with MM on the face and neck may be linked to the concept of two routes of MM exposure to sunlight. Traditionally, an increased incidence of MM has been associated with individuals who have intense intermittent UV radiation exposure and multiple solar nevi. However, another route of UV influence has been proposed for patients with the histological subtype lentigo maligna-MM, which is predominantly located on the face and neck. In these cases, chronic occupational exposure to UV radiation may contribute to an increased risk of developing SPN [30]. This theory suggests that the etiology and pathogenesis of MM may vary depending on the histological type, and lentigo maligna mm may share a common carcinogenesis process with other SPNs observed in our cohort.

The high percentage of skin tumors, particularly NMSC, reported in our cohort further supports the role of UV radiation in the increased risk of SPN in our population. Additionally, other studies have reported different risk factors for developing a second MM, such as fair hair color, more than 100 common melanocytic nevi, or the presence of more than 50 cherry angiomas. However, these studies did not find an increased risk based on the site or histological subtype of the first MM [3].

Overall, genetic predisposing variants are polymorphic and have different levels of penetrance what implies a risk just for a small percentage of the population. However, environmental factors, such as toxics exposure, lack of access to surveillance programs, and regarding MM, UVR, are commonly distributed among populations, and different duration and intensity exposition rates could explain the variety in results and SPN associations reported to date.

Finally, we consider important to emphasize that our study revealed a significantly elevated risk for SPNs among patients aged over 50 years, with a history of lentigo maligna-MM, and with MM located in the head and neck region. As a result, the robustness of our findings supported the recommendation for more frequent and thorough monitoring in this high-risk group among our population.

### Limitations

Despite the efforts of the authors, the study poses a series of limitations that must be considered. It is possible that a selection bias has been committed, since after a diagnosis of MM, patients tend to have a healthier lifestyle, reducing the impact of possible environmental factors associated with the appearance of malignant neoplasms. This could lead to underestimation of the risk in our cohort compared to the general population. In addition, the short follow-up of the patients (3.8 years on average) makes it possible that neoplasms developed over a longer period of time have not been detected. The role of surveillance bias in the detection of SPN has been commented on previously, and appeared to not have a strong influence on the observed results. Finally, the lack of quantified data for variables such as sun exposure or the number of moles did not allow us to analyze these phenotypic and environmental characteristics related to the pathogenesis of MM.

**Figure 1 cancers-15-03056-f001:**
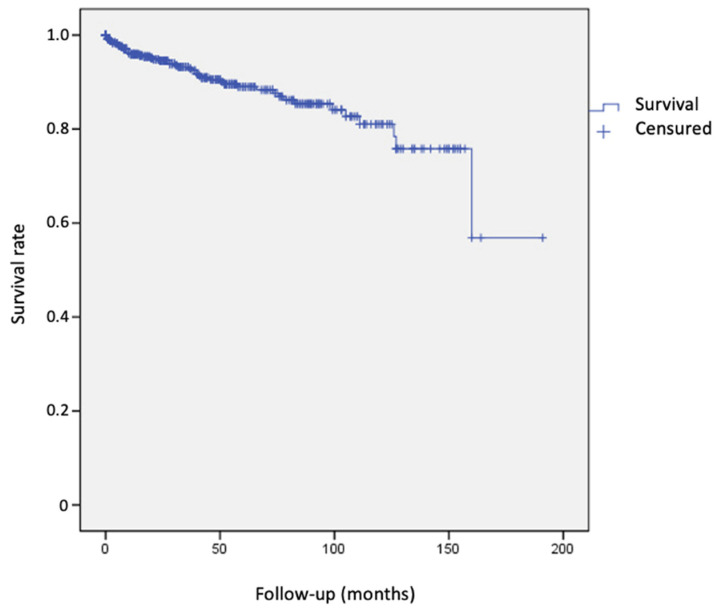
Survival curve for the development of second primary neoplasms. The risk increased over time, reaching its peak after 150 months of follow-up.

**Figure 2 cancers-15-03056-f002:**
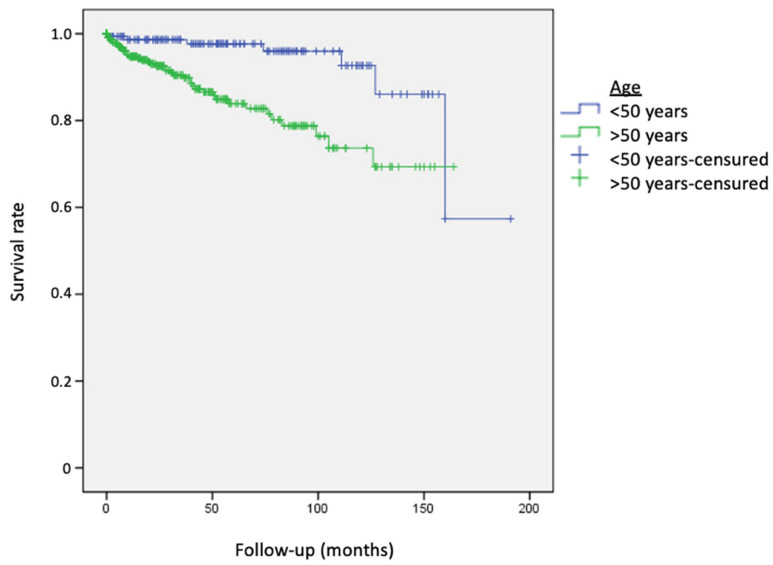
Survival curves for the development of second primary neoplasms in patients older and younger than 50 years. Patients “older than 50 years old” (green line) presented second primary neoplasms more easily and earlier. Log-rank χ^2^ = 13.31 DL = 1, *p* < 0.0001.

**Figure 3 cancers-15-03056-f003:**
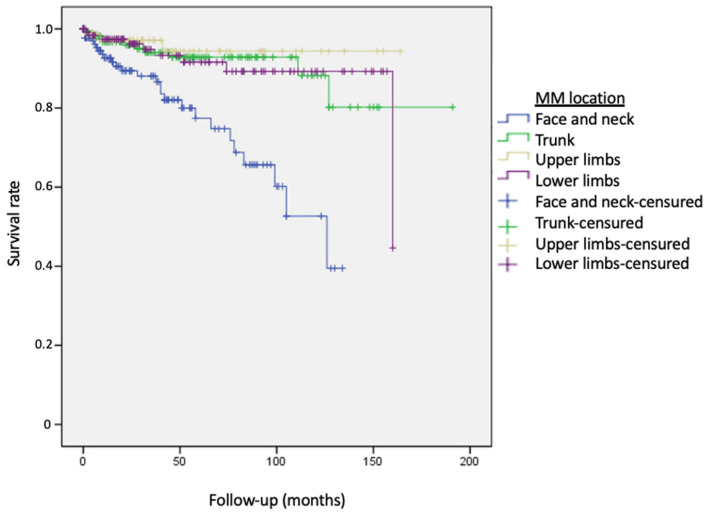
Survival curves for the development of second primary neoplasms, according to location of primary melanoma. Patients were more likely to present second primary neoplasms if the previous melanoma was located on the face and neck (blue line). Trunk (reference face and neck) Log-rank 15.75 *p* < 0.0001. Upper limbs (reference face and neck) Log-rank 15.75 *p* < 0.0001. Lower limbs (reference face and neck) Log-rank 15.75 *p* < 0.0001.

## 5. Conclusions

In conclusion, our study identified an increased risk of developing SPNs only in a subgroup of MM survivors, particularly in those over 50 years old, with primary MM located on the face and neck, and with lentigo maligna mm as the histological subtype. Understanding these risk factors has significant implications for our population in terms of follow-up care, the development of specific guideline recommendations, and the exploration of new genetic or environmental pathways involved in MM. Further studies with a larger number of patients and longer follow-up periods are needed to establish an etiopathogenic model that elucidates the epidemiological link between MM and the occurrence of SPNs.

## Figures and Tables

**Table 1 cancers-15-03056-t001:** Characteristics of the melanomas in the study cohort. The most common characteristics observed were superficial spreading melanoma as the histological subtype and the head and neck region as the most frequent location.

Histological Subtype	
Lentigo maligna MM	63 (11.9%)
Superficial spreading MM	213 (40.3%)
Nodular MM	192 (36.3%)
Acral lentiginous MM	26 (4.9%)
Non-cutaneous MM	35 (6.6%)
MM location	
Face and neck	142 (27.5%)
Trunk	172 (33.3%)
Upper limbs	76 (14.7%)
Lower limbs	126 (24%)
Breslow index (mm)	
Mean (SD)	2.67 (3.9)
Range	0–44
Clark level	
I (Intraepidermic)	63 (12.8%)
II (Papillary dermis)	106 (21.5%)
III (Medium dermis)	159 (32.3%)
IV (Reticular dermis)	116 (23.5%)
V (Hipodermis)	49 (9.9%)

**Table 2 cancers-15-03056-t002:** Characteristics of patients with second primary neoplasms found in our population. Out of the total 54 s primary neoplasms identified, 34 (63%) were skin tumors and 20 (37%) were solid organ tumors.

		Second Primary Neoplasms	
	Total	Skin Tumors	Solid Organ Tumors *
No. of patients	529	34 (6.4%)	20 (3.8%)
Gender			
Male	240 (45.4%)	18 (52.9%)	12 (60%)
Females	289 (54.6%)	16 (47.1%)	8 (40%)
Age of diagnosis (years)			
Mean (range)	60 (11–94)	66.4 (35–94)	63.6 (30–82)
<50 years (n)	157 (29.7%)	6 (17.6%)	2 (10%)
>50 years (n)	372 (70.3%)	28 (82.4%)	18 (90%)
Age of diagnosis of SPN (years)			
Mean (range)	67.5 (35–94)	69.1 (35–94)	68.4 (44–89)
Follow-up (months)			
Mean (range)	45.4 (0–191)	32.8 (1–127)	55.8 (3–160)

* Solid organ tumors observed: 3 breast carcinomas (15%), 6 digestive tract tumors (30%), 3 prostate carcinomas (15%), 1 laryngeal carcinoma (5%), and 7 tumors from other locations (35%).

**Table 3 cancers-15-03056-t003:** Estimated cumulative probability of developing a second primary neoplasm in patients with malignant melanoma. The risk was higher in the “older than 50 years” group, and this risk increased with advancing age.

	1 Year	5 Years	10 Years
Global	4.1%	11%	19%
Age ≤ 50 years	1.4%	2.3%	7.3%
Age > 50 years	5.3%	16.1%	26.3%
Solid organ tumors	3.2%	8.1%	12.1%
Age ≤ 50 years	1.4%	2.3%	7.3%
Age > 50 years	4%	11.5%	14.3%
Solid organ tumors	1.1%	3.4%	9%
Age ≤ 50 years	0	0	0
Age > 50 years	1.7%	7.3%	15.9%

**Table 4 cancers-15-03056-t004:** Results of the univariate and multivariate analyzes using the Cox regression model. Being over than 50 years of age, having a history of melanoma located on the head and neck region, and having lentigo maligna melanoma as the histological subtype, were the only variables statistically significant associated with an increased risk of developing second primary neoplasms.

Variables	Univariant Analysis Rr (Ic 95%)	*p* Value	Multivariant Analysis Rr (Ic 95%)	*p* Value
Male gender	0.6 (0.34–1.05)	NS	^a^	
Age (years)	1.05 (1.03–1.07)	<0.0001	1.04 (1.02–1.07)	<0.0001
Breslow index (mm)	1.04 (0.97–1.11)	NS	^a^	
MM location:				
Reference group: Face and neck				
Trunk	0.26 (0.13–0.54)	<0.0001	0.24 (0.11–0.54)	<0.001
Upper limbs	0.17 (0.05–0.55)	0.003	0.13 (0.04–0.45)	<0.001
Lower limbs	0.27 (0.12–0.58)	0.001	0.20 (0.08–0.52)	<0.001
MM histological subtype:				
Reference group:				
Superficial spreading MM	0.13 (0.86–3.15)	NS	0.94 (0.46–1.89)	NS
Nodular MM	0.27 (0.68–3.95)	NS	0.30 (0.11–0.81)	0.018
Lentigo maligna MM	0.23 (0.62–7.32)	NS	1.54 (0.37–6.43)	NS
Acral lentiginous MM	0.89 (0.26–4.84)	NS	0.31 (0.07–1.45)	NS

Likelihood ratio test = 46.598; *p* = 0.0000; DL = 8; RR: relative risk; IC 95%: 95% confidence interval. NS: non-significative; ^a^ Deleted from multivariant analysis (*p* > 0.15).

## Data Availability

Supporting data cand be provided by contacting with the corresponding author.

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
