# Peer review of "Risk of Second Primary Malignancies in Melanoma Survivors: A Population-Based Study"

_cancers, 2023, doi:10.3390/cancers15113056_

Round 1

Reviewer 1 Report

This interesting report reviews the development of second neoplasms in patients with MM. Design is correct and conclusions are sound, although I feel there is no real increase  in the incidence of malignant tumors in this population, but rather an increase in survival associated to therapy improvement, that can lead to the development of the most common malignancies in general population, as breast or GI carcinomas. I think this should be discussed further in the Discussion section of the manuscript. I do not think there is a causal relationship, but rather a longer survival and a closer up surveillance. It would be interesting to know whether most tumors were dectected in an earlier stage.  Besides, I would like to read some information regarding which patients should be more intensively followed according to their results 

English is good win minor mistakes

Author Response

We would like to express our sincere gratitude for the opportunity to have our work reviewed and for your invaluable suggestions for improvement. Your insightful comments and constructive feedback have been thoroughly considered, leading to significant enhancements in the manuscript.

Throughout the review process, we carefully incorporated all of your recommendations into the revised version of the manuscript. We are pleased to report that these modifications have resulted in a new version that is more coherent, precise, and of higher quality overall.

We genuinely appreciate the time and effort you dedicated to critically evaluating our work. Your expertise and attention to detail have undoubtedly contributed to the substantial refinement of our research. We are grateful for your valuable contributions, which have significantly strengthened the clarity and scientific rigor of our findings.

Reviewer 1

This interesting report reviews the development of second neoplasms in patients with MM. Design is correct and conclusions are sound, although I feel there is no real increase  in the incidence of malignant tumors in this population, but rather an increase in survival associated to therapy improvement, that can lead to the development of the most common malignancies in general population, as breast or GI carcinomas. I think this should be discussed further in the Discussion section of the manuscript. I do not think there is a causal relationship, but rather a longer survival and a closer up surveillance. It would be interesting to know whether most tumors were dectected in an earlier stage.  Besides, I would like to read some information regarding which patients should be more intensively followed according to their results 

Response

We totally agree with your suggestion and the following text has been added to the discussion section:

In our study, we did not observe a significant overall increase in the risk of SPNs among patients with a history of MM when compared to the risk in the general population. The mean age of our patients was 60 years, and we observed a prevalence of 16.8% for SPNs, with 11% of them occurring after the diagnosis of MM. These results were very similar to the risk of developing any neoplasm in Spain at that age, which is 15%.

Furthermore, when excluding skin tumors, the incidence of the most commonly observed solid tumors, such as gastric and breast cancers, exhibited similar or even lower rates compared to the national average (2% vs. 1% and 2% vs. 8%, respectively). Thus, it is important to note that these findings do not completely eliminate the potential risk for long-term surveillance bias. Moreover, it can be suggested that the solid SPNs identified in our series, could be detected due to the close follow-up of patients with a history of MM, rather than indicating an overall increased risk in this population. Consequently, solid SPNs were identified at earlier stages with all the tumors being discovered at stages I and II, with no cases of distant disease.

Nevertheless, we consider important to emphasize that within our study, a significantly higher risk of SPNs was observed among patients older than 50 years, with a history of lentigo maligna-MM, and with MMs located in the head and neck region. Thus, we conclude that this specific group have a real higher risk and should receive more frequent and thorough monitoring.

Reviewer 2 Report

General comments to the authors

n  The authors address the subject of “Risk of second primary malignancies in melanoma survivors” (A population-based study)

Specific comments

n  The rationale behind the study should be clearly stated throughout the manuscript at the start of the "Abstract section", and at the end of the “Introduction section”.

n  The authors should provide some more details about the statistical analysis. This section should be more developed in the revised version of the manuscript.

n  The main findings of the study should be summarized at the start of the “Discussion” section. Then each important should be discussed.

n  The pathogenetic mechanisms underlying the risk of second malignancy in melanoma survivors should be discussed (Discussion section).

n  A comment (section) addressing the “Strengths” of the current study. In other words, what are the new insights that this current study added to our current knowledge about this issue (Risk of second primary malignancies in melanoma survivors)?

n  What are the research questions and avenues that the current study opens for future investigations?

n  The authors should add a short comment (Figure and table legends) following each table and figure to summarize the salient findings. As they stand, all the figures and tables are not informative by themselves.

Minor editing is required.

Author Response

We would like to express our sincere gratitude for the opportunity to have our work reviewed and for your invaluable suggestions for improvement. Your insightful comments and constructive feedback have been thoroughly considered, leading to significant enhancements in the manuscript.

Throughout the review process, we carefully incorporated all of your recommendations into the revised version of the manuscript. We are pleased to report that these modifications have resulted in a new version that is more coherent, precise, and of higher quality overall.

We genuinely appreciate the time and effort you dedicated to critically evaluating our work. Your expertise and attention to detail have undoubtedly contributed to the substantial refinement of our research. We are grateful for your valuable contributions, which have significantly strengthened the clarity and scientific rigor of our findings.

Reviewer 2

The authors address the subject of “Risk of second primary malignancies in melanoma survivors” (A population-based study)

Specific comments

     1.- The rationale behind the study should be clearly stated throughout the manuscript at the start of the "Abstract section", and at the end of the “Introduction section”.

Response

We totally agree with your suggestion and the following text has been added to start of the Abstract section and to end of the Introduction section:

      The aim of this study was  to evaluate the parameters that in our population are associated with a greater risk of SPNs after a history of MM. The knowledge of these risk factors would make it possible to establish follow-up protocols with specific recommendations for individuals with the highest risk.

   2.- The authors should provide some more details about the statistical analysis. This section should be more developed in the revised version of the manuscript.

Response

      Based on your suggestion, we have added some additional explanations and details to the Statistical Analysis section.

       3.-The main findings of the study should be summarized at the start of the “Discussion” section. Then each important should be discussed.

       Response

       Now, we highlight the most important findings of our article at the beginning of the discussion section. Among our population, patients who were older than 50 years, had a history of lentigo maligna-MM, and had MMs located in the head and neck region exhibited the highest risk. Therefore, we strongly recommend more frequent and comprehensive monitoring for this specific group.

     4.-The pathogenetic mechanisms underlying the risk of second malignancy in melanoma survivors should be discussed (Discussion section).

       Response

We totally agree with your suggestion and the following text has been added to the Discussion section:

       The risk of developing SPNs in MM survivors can be influenced by various pathogenetic mechanisms, including genetic, environmental, treatment-related and shared risk factors. For example, mutations in the BRAF oncogene, commonly found in MM, have also been observed in other cancers such us cell leukemia, papillary thyroid, colorectal, liver, brain, lung, ovarian or breast cancer. Moreover,  germline mutations in BRCA2 or CDKN2A have been proposed as potential explanations for the association between MM and breast or pancreatic cancer. Environmental factors, such as exposure to ultraviolet radiation (UVR) from sunlight, are other well-established contributors to both primary and second MM. Additional factors like tobacco use, occupational exposures, and socioeconomic status (SES) can also contribute to the risk. Treatment-related factors, such as chemotherapy or radiotherapy used for primary MM treatment, may increase the risk of developing SPNs. Lastly, certain life styles can behave as shared risk factors, such as obesity or lack of physical activity, and may explain the occurrence of multiple malignancies in the same individual.

    5.-  A comment (section) addressing the “Strengths” of the current study. In other words, what are the new insights that this current study added to our current knowledge about this issue (Risk of second primary malignancies in melanoma survivors)?

       Response

We totally agree with your suggestion and the following text has been added to the end of the Discussion section:

We consider important to emphasize that within our study, we observed a significantly higher risk of SPNs among patients older than 50 years, with a history of lentigo maligna MM, and with MMs located in the head and neck region. Thus, we conclude that this specific group have the highest risk and should receive more frequent and thorough monitoring.

     6.- What are the research questions and avenues that the current study opens for future investigations?

       Response

      This study identifies the variables that contribute to a higher risk of SPNs in our population, providing valuable insights for establishing follow-up recommendations. However, we consider  crucial to validate and replicate these findings in diverse samples, as the distribution of genetic variants and environmental factors influencing risk may differ among populations. Furthermore, our article serves as a catalyst for future research endeavors with larger patient cohorts and longer follow-up durations, which are necessary to establish an etiopathogenic model that elucidates the true epidemiological connection between MM and the occurrence of SPNs.

     7.- The authors should add a short comment (Figure and table legends) following each table and figure to summarize the salient findings. As they stand, all the figures and tables are not informative by themselves.

       Response

We totally agree with your suggestion and a short explanation has been added following each table and figure:

Table 1. Characteristics of the melanomas in the study cohort. The most common characteristics observed were superficial spreading-melanoma as the histological subtype and the head and neck region as the most  frequent location.

Table 2. Characteristics of patients with second primary neoplasms found in our population. Out of the total 54 second primary neoplasms identified, 34 (63%) were skin tumors, and 20 (37%) were solid organ tumors.

Table 3. Estimated cumulative probability of developing a SNP in patients with malignant melanoma. The risk was higher in the "older than 50 years" group, and this risk increased with advancing age.

Figure 1. Survival curve for the development of second primary neoplasms. The risk increased over time, reaching its peak after 150 months of follow-up.

Figure 2. Survival curves for the development of second primary neoplasms in patients older and younger than 50 years. Patients “older than 50 years old” (green line) presented second primary neoplasms more easily and earlier.

Figure 3. Survival curves for the development of second primary neoplas,s according to location of primary melanoma. Patients were more likely to present second primary neoplasms if the previous melanoma was located on the face and neck (blue line).

Table 4. Results of the univariate and multivariate analyzes using the Cox regression model. Being over than 50 years of age, having a history of melanoma located on the head and neck region, and having lentigo maligna-melanoma as the histological subtype, were the only variables statistically significant associated with an increased risk of developing second primary neoplasms.